# Learning Multiple Movements in Parallel—Accurately and in Random Order, or Each with Added Noise?

**DOI:** 10.3390/ijerph191710960

**Published:** 2022-09-02

**Authors:** Julius B. Apidogo, Johannes Burdack, Wolfgang I. Schöllhorn

**Affiliations:** 1Akanten Appiah-Menka University of Skills Training and Entrepreneurial Development, Kumasi AK-039, Ghana; 2Department of Training and Movement Science, Institute of Sport Science, Johannes Gutenberg-University Mainz, 55099 Mainz, Germany

**Keywords:** underarm pass, overhand serve, overhand pass, differential learning, parallel learning, sequential learning, multiple skills, contextual interference, volleyball

## Abstract

Traditionally, studies on learning have mainly focused on the acquisition and stabilization of only single movement tasks. In everyday life and in sports, however, several new skills often must be learned in parallel. The extent to which the similarity of the movements or the order in which they are learned influences success has only recently begun to attract increased interest. This study aimed to compare the effects of CI in random practice order (high CI) with differential learning (DL) in learning three volleyball skills in parallel. Thirty-two advanced beginners in volleyball (mean age = 24, SD = 2.7) voluntarily participated in the study. Within a pre-, post-, retention test design, an intervention of six weeks and one week retention phase, the effects of three practice protocols of a CI, DL, and control (CO) group were compared. Three different volleyball skills (underhand pass, overhand pass, and overhand serve) were trained with emphasis on accuracy. Results showed statistically significant higher rates of improvement in the acquisition and learning phases for the DL group compared to the CI and CO groups. The differences were associated with moderate to high effect sizes in all individual skills and in the combined skills. The findings show more agreement with DL than with CI theory.

## 1. Introduction

In times of ever faster change and increasing automation of everyday working life, mental and physical flexibility together with the associated motor learning are becoming more and more important. In this context, it is not only what is learned that is of interest, but also the “how” of learning in its entire scope is increasingly attracting interest. Despite the practical experiences and reports of coaches and physical education teachers on the mutual influence of learning multiple movements in parallel, and especially when learning two very similar movements, the subject is widely neglected in research. The problem arises, for example, when the backward somersault and the backhand spring are learned at the same time in gymnastics, or when the rotational shotput skill and the discus throw rotation are learned in parallel in track and field. In psychomotor research, the problem is partially approached with the paradigm of pro- and retroactive inhibition [1]. In connection with sports hitherto, only two motor learning approaches addressed the issue of learning multiple skills in parallel scientifically, the contextual interference (CI) [2,3] and the differential learning (DL) approach [4,5].

In the beginning, the CI approach was launched by adding additional exercises (=con-text) to the learning of a single movement that had to be executed between the repetitions of the actual to-be-learned (=text) movement [6]. Thereby, a randomized sequence of exercises during the acquisition phase was often accompanied by two phenomena, drawbacks (=interference) during practice followed by advantages in the later conducted retention tests. Regarding the concept of the acquisition phase, however, it must be kept in mind that all the CI-studies did not actually involve the acquisition of movements in the sense of learning a completely new movement, but rather exclusively the stabilization of discrete movements that were mastered from the beginning and which were only executed faster towards a given target or with fewer errors at the end. Nonetheless, with the introduction of the additional testing of the exercises that were originally introduced as context, the focus increasingly shifted towards the parallel learning of several movements. Regardless of this transfer and the recommended randomized practice sequence, the movements were always precisely defined and had to be performed correctly and according to the description of a prototype. Erroneous movements during the stabilization phase had to be repeated to ensure the same number of correct movements in all groups [7]. After initial evidence for movements with a small number of degrees of freedom (sDGF) in young adults [3,8], the CI-approach has meanwhile increased evidence for limited generalization, particularly in four areas. There is increasing doubt about the transferability of the CI model (a) to movements with large degrees of freedom (lDGF) [9,10], (b) to learning in children [11] (c), to other than the stabilization phase or highly skilled learners [12,13], and (d) to other than mainly sequential tasks with dominant visual–spatial content [14]. The extent to which these aspects are an independent phenomena remains to be clarified.

Scrutinizing the CI-related studies on volleyball (Table 1) [15,16,17,18,19,20,21,22,23] support the mentioned restrictions. If only all volleyball studies in the context of CI studies with a comparison between random and blocked schedule are concerned, the first thing to notice is that only the two studies with young adults [19,21] show both CI phenomena (marked with two Y in the last two columns of Table 1). None of the studies with juveniles or adolescents showed the interference phenomenon immediately after the practice phase or improved retention results for the randomized practicing groups. Only in two studies the randomized-exercising group outperformed the blocked-exercising control groups on transfer tests [16,22].

Within the first generalization attempts, the “variability of practice” approach [24], which relies on Schmidt’s schema theory [25], was interpreted as a subset of the CI approach. Varying variable parameters, such as absolute forces or absolute timing within the same movement with the same invariants, such as relative forces or relative timing (same generalized motor program (GMP)), became distinguished from varying between different movement skills, with the larger effects attributed to the second [26]. Structurally, they indicated together that the movements to be learned in parallel should not be too similar, keeping in mind that the variability of practice model is only applicable to movements without gravitational and without inertial forces [14]. However, quantitative research on the extent to which the relative similarity of the movements to be learned in parallel or the size of the parameter differences influences the learning process has so far been out of focus. Momentarily, from the CI theory point of view, it would be no difference if a tennis service, a javelin throw, and a volleyball service are learned in parallel or a tennis service, a volleyball underhand pass, and a football kicking movement [27]. Both should result in the same phenomena for all motor skills because the number of exercises with different GMPs is the same. Furthermore, it is unclear, whether the phenomenological similarity is the critical criteria, or it is the similarity of the underlying biomechanical force characteristics in connection with the neurological control mechanisms or some other dimension. This neglect is somewhat perplexing since the influence of activities performed immediately before and immediately after the learning process was already in the focus of research in the early days of CI research on learning movements within the pro and retroactive inhibition paradigm [28,29,30,31].

For explaining the first CI-effect, the interference phenomenon, the model of an overload of the working memory was suggested that could be caused by an increased number of exercises to be remembered [3], by time pressure [32], by frustration [33], or by increased processing due to previous movement errors [34]. While these explanations are still plausible for acute effects in experiments with a short duration, e.g., [35], when the practice phase lasts only one day and the post-test is performed at the end of the practice phase, this explanation becomes problematic for experiments that extend over several weeks, as is observed in most CI-studies on sports movements. This also could be a reason for the frequent absence of an interference effect in movements with many degrees of freedom when the practice phase lasted several weeks [9]. Regardless of this, all these suggestions rely on a working memory model that had been suggested for tasks that were sequential in their characteristic with mainly language or visual–spatial content but not for tasks with dominant kinesthetic, proprioceptive, or tactile influence and parallel processing [14,36]. An alternative explanation model for the interference phenomenon is provided with the forgetting model [37,38]. The forgetting model assumes a time dependent decay of traces of the to be learned action plan during the execution of the other movements. In difference to the idea of overloading the working memory the forgetting model is not only dependent on the number of additional exercises but also on the time between two repetitions. Thus, the model additionally includes the possibility of a gradual change of the learning system leading to the altered perception and execution of the same movement with continuously changing conditions. This in turn would increase the variety of experiences in the surrounding of the solution and at the same time provide an explanation for the second phenomenon.

For the second CI phenomenon, the increased retention performance, besides the forgetting model, two models from cognitive psychology were suggested that are rather related to the long-term memory. The elaboration model [3] assumes more elaborative processing of task-related information during the random sequence and leads to advantages in the retention test. The reconstruction model [39] sees the cause in the constantly new construction of the to be learned action plan(s). While both models argue with two separated forms of memory and miss an explanation for its transition, the forgetting model is applicable to both phenomena. Only in case of complete forgetting would the forgetting model become identical with the reconstruction model. Interestingly, the reconstruction model explicitly pursues the classical consolidation of the action plan in the long term memory by repetition, whereas the other two models structurally already take up a topic which was indicated by Bernstein’s well-known quotation “repetition without repetition” [40], but only found its establishment with the more recent findings about the functioning of natural and artificial neural networks (ANN) [41] and experienced practical realization in the differential learning approach.

Indirect support by means of analogies from neurophysiological studies is mainly provided for the elaboration and reconstruction models [42]. Unfortunately, scrutinizing the forgetting hypothesis and its bearing is hitherto missing. Irrespective of this, due to the primary application of MRI measurement, the movements had sDGF and large participation of the visual sense. Mainly an increase in activity during the execution of random sequences of exercises has been found in cortical areas that are typically associated with working memory [43] and provide evidence for the hypothesis of a coarse working memory overload. Regardless of the increasingly frequent confirmed limitations of the CI model in the meantime, the approach has generally stimulated learning research in terms of the distinction between a practice and a retention phase, as well as in terms of rethinking the structure of practice sequences. Instead of trying to squeeze every foot into the same shoe, it would probably be more promising to use the confirmed and useful elements for new shoes in the future.

The differential learning approach (DL), based on system dynamics (SyDy) and artificial neural net (ANN) research [5,44], counterintuitively treats learning as an aspect of living systems that have been largely neglected. While the program-oriented models considered deviations from an ideal as destructive elements and errors to be avoided, in SyDy and ANN research they were constructive and essential elements of the theory. While in SyDy deviations were treated as necessary fluctuations or noise of living systems whose amplitude made decisions about stability and instability [44], in ANN research it was the noisy training of the networks that led to powerful applications [41]. ANNs are learning worse when the training data are too similar. They cannot interpret data that are more deviating [41]. Consequently, the ANNs, whose original architectures were derived from real neuronal networks, are trained with noisy data in a broader solution space to allow interpolation during application phase and avoid extrapolation [4,45]. In SyDy of living systems a transition from one stable state to another is characterized by an increase in fluctuations that make the systems unstable, which needs less energy to change the momentary state and initiates a self-organizing process in the sense of providing diffuse energy without explicit information about the structure of a possible solution, even not in form of identifying errors (which includes explicit information for the solution as well). Instead of waiting passively until the learning system is moving towards an instability, noisy learning in the DL approach became realized by actively adding stochastic perturbations to the movement to be learned or by actively amplifying existing fluctuations. Thereby the stochastic perturbations can have external and internal origins [45]. The interaction between the exercise’s noise, mostly given from the coach’s chosen teaching approach, and the athlete’s response noise was described by the metaphor of stochastic resonance, which only results in maximum performance when both noises are in resonance [4,45,46]. Thereby the optimum noise varies with the individual and its situation, which is dependent on his or her emotions, fatigue, daily changes, etc. [47,48,49,50,51,52].

In contrast to the psychologically based approaches the DL-approach began with studies on complex sports movements [53,54,55,56] before studies on everyday movements [57] and movements with sDGF [58,59] also supported the predictions that were made in the beginning [4,5]. Similar to the historic development of the CI approach, DL research started with studies on the acquisition and stabilization of single movements before the approach was transferred to the parallel learning of two and three skills [60,61,62].

For explaining the phenomena, DL theory also assumes an overload of working memory during the exercise phase, but in contrast to CI theory, it assumes a qualitative switch in information processing already during the exercise process by getting rid of the controlling and limiting activities of the frontal lobe to be able to use additional neuronal resources [63,64]. This is reflected in an increase in lower EEG frequency bands in areas that are typically associated with working memory [65,66]. Lower frequencies, especially within the alpha band, go along with the integration of several cortical areas [67], whereas the higher gamma- and beta frequency bands are rather associated with local and highly intensive processing [64]. The switch during a DL exercise process is attributed to different neural processing of visual and kinesthetic proprioceptive information, which show different dominance in CI and DL experiments [14]. Whether the switch depends on the quality or quantity of processed information is still unanswered.

Due to the forthcoming DL-approach and due to the philosophy of multiple arguments instead of a large number-statistics [14], a much smaller number of studies on volleyball have been conducted. First, DL-studies on volleyball analyzed the stabilization of underarm passing skill in adolescents comparing open metaphorical movement tasks that constantly changed with corrective and model oriented feedback [53]. After 4 weeks, both groups improved their precision with underarm passing, but the DL group statistically significantly outperformed the model-oriented group. Similar results were obtained in a three-week intervention study [68]. The stabilization of two volleyball skills, the underarm and overhand pass, by means of the DL approach in a group of 51 mixed pupils was investigated in another study [61]. In both skills, the DL group statistically significantly outperformed the repetitive group in post-, retention, and transfer tests. The constantly training group was only able to improve the performance in one volleyball skill, no changes in performance could be diagnosed for the second volleyball skill, indicating a mutual interaction for this type of learning. In contrast to previous studies on single skills, the DL group could also improve both skills from pre to post-test but only one skill improved even further until the retention test, while the second skill only kept the performance level of the post-test.

Most recently, the DL approach was tested on the simultaneous acquisition of the three basic volleyball skills in female juvenile beginners and compared with a repetitive approach as well as with a group that pursued a general ball habituation strategy [62]. After 6 weeks of intervention only the DL group achieved statistically significant improvements in all three volleyball skills in post- and retention tests whereas the other two groups only improved both passing skills but not the service and much less than the DL group. One could interpret this as a refutation of the repetition approach. However, a more detailed analysis of the origin of the repetition model reveals that this model was originally designed neither for the simultaneous learning of several movements nor for use with adolescents, but rather generalized into different dimensions due to a lack of alternatives and inadmissibly from a scientific-theoretical point of view [14].

This study aims to compare the DL approach with an approach that also focuses on the simultaneous learning of multiple volleyball skills. Therefore, the objective of the study is to compare the effects of the CI and DL approaches on the simultaneous learning of the three basic volleyball skills in young adults. The comparison focuses on outcome analysis in the form of hitting rates and omits analysis of the movement process.

Following the theory of the CI approach, we expect both the CI and DL groups having a lower performance level after the exercise phase than the general skill-based training group due to the high number of different skill-specific exercises and the accompanied working memory overload. Additionally, since the DL group has significantly fewer correct repetitions of the target exercises, according to the CI theory, the DL group should perform worse than the CI group. Due to the interference during the stabilization phase, both DL and CI groups are expected to outperform the control group in the retention test. No differences between CI and DL group are expected in the retention test.

In contrast, according to the DL theory, the DL group should outperform all other training groups already after the practice phase due to the reduced activity of the controlling prefrontal cortex and the more differentiated coverage of the possible solution spaces of all three techniques on a neural network level. Due to the downshift of control by the prefrontal cortex [68] and the accompanying increased activations for consolidation in areas of the motor cortex [65,66], better performances are also expected in the retention tests in the DL group than in the other two groups.

## 2. Materials and Methods

### 2.1. Participants

A total number of 36 University of Education students, 33 men 3 women, who were advanced beginners in volleyball (Table 2) voluntarily participated in the study. After they had been explained the content and purpose of the study, all participants gave their informed written consent. The procedures were carried out in accordance with the Helsinki declaration guidelines and approved by the institutional review Board of Akenten Appiah-Menka University of Skills Training and Entrepreneurial Development (AAMUSTED/RO/L.3/361 10 March 2022).

### 2.2. Design

The study adopted the design by [48]. The groups for this study were contextual interference group (CI), a differential learning group (DL) and a control group (CO). 12 participants were in each group.

#### 2.2.1. Intervention

The study was conducted following the design of [61] with three intervention groups whose performances were assessed in pre-, post- and retention tests. The CI group trained three exactly prescribed skills in a random order (high CI) in line with Battig [2,6]. The intervention period lasted for six weeks. Participants trained thrice a week (Mondays, Wednesdays, and Fridays) which resulted in 18 training session overall. Each training session was two hours (2 h). All training sessions were afternoons. Each training session was preceded by a warmup activity for 10 minutes and activities were minor games such as ‘’three against one”, ‘’chase and catch”, and ‘’one against seven’’. After the warmup activities, the subjects went to practice in their various groups and closed after that. The CI group’s training was characterized with practicing the three skills randomly per session, making 20 attempts at each skill from overhand service (S) to overhand pass (V) volleying) to underhand pass (D) (e.g., SDSVDVSDDVSDVV…) following that order repeatedly per session. The underarm and overhand pass skills were practiced with a toss of the ball from a trained person for that purpose, while the overhand serve was self-tossed. The DL group practiced according to the recommendations of [5], starting with variations mainly in the geometry of movement and posture as it was applied in [61]. All participants received constantly changing movement instructions randomly selected before each training session from the list of exercises displayed in Appendix A. The exercise list is mainly characterized by adding noise to the motor skill being learned through joint-related variations [5]. In all training sessions, no exercise was repeated twice and no corrections or augmented feedback were given to initiate a true self-organizing process in which no explicit information about the solution is given, neither related to incorrect or correct executions. Each participant in the DL group had 20 variants per skill per training session for a six week acquisition period, 180 per participant a week and 1080 trials in all for the entire period for both CI and DL groups. The control group was engaged in non-volleyball games such as playing minor games which were not volleyball related, example throwing and catching ball games, aiming targets with a ball, they did nothing specific concerning volleyball.

#### 2.2.2. Test Design

At the end of the sixth week, we conducted a post-test followed by a retention test one week later. In all tests the subjects performed 10 trials in each skill and each test in balanced sequence. Therefore, the maximum score for each test was 40 points and minimum of zero which was given if the ball did not land inside the target and did not touch any of the lines of the marked target areas.

The balls were thrown to participants by a human ball feeder who was trained in it for several weeks before the investigation started. The rule for the test for the thrown ball to participants was that the ball must reach the participant at a static position within a marked area in which they were standing, for them to execute the skill. If they had to move out of that position to perform the skill, it was not counted as part of the 10 trials and had to be repeated. The test was conducted in blocked order from the underarm pass to overhand pass to overhand serve. Each test was conducted and concluded the same day under similar conditions.

The American Association of Health Physical Education and Recreation (AAHPER) volleyball skill test was used to collect data. There were three skills involved in the test: the underhand pass, overheard pass, and overheard serve. The tests were carried out on a regular volleyball court size of 18 by 9 m.

Sub-test underhand pass (Figure 1A): To test the underhand pass accuracy, the student stood in a 2 m^2^ square on the right-hand side of the volleyball court (zone Z5) and received a ball thrown from zone 2 of the other court and passed the ball over a rope (height 2.34 m) into a 3 m × 2 m target area in zone Z2 of the participant’s court for which 4 points were awarded if the ball lands in the target area and 2 points were awarded if it lands on the lines of the target area.

Sub-test overhand pass (Figure 1B): To test the accuracy of the overhand pass, the participant stood in zone Z2, received the ball from zone Z6, and passed the ball over a 2.34 m high rope into two 1 m × 2 m target areas, with the one farther from the participant scoring 4 points, the one closer scoring 2 points, and the line in between scoring 3 points.

Sub-test overhand service (Figure 1C): The participants stood at the end of the field in a central 2 m wide area and served the ball over their head to the other field over the 2.34 m high net into the 2 m × 2 m rectangular target areas, with points awarded for each area. The further back and sideways the target area that was hit, the more points a serve resulted in, ranging from 1 to 4 points. In between the zones, the two zone points were added and divided by two and the points were given.

For the multiple skills test, the scores of all three single tests were combined by means of z-transformed data.

Six research assistants were trained to assist in the process of training and conducting the test. The execution of an attempt was counted only if the ball thrown by the research assistant was receivable by the participant within the marked area. Otherwise, the attempt was repeated.

### 2.3. Data Analysis

The groups were compared statistically based on their results in each skill and in combined multiple skills. To check the internal consistency of the tests for the respective skills, five participants each performed the respective test at intervals of one week. Cronbach’s alpha was determined based on the values from weeks 1 and 2. Analyses of the data using Shapiro–Wilk tests revealed that some variables violated the assumption of normal distribution. Consequently, the development of the groups across the measurement time points and the comparison of the groups at the respective measurement time points were performed using non-parametric statistical tests. For the analysis of the development within the groups in the respective skills at pre-, post-, and retention test, the results of the tests were statistically compared using Friedman ANOVA. In case of significant results, pairwise Bonferroni-corrected post-hoc Dunn–Bonferroni tests were performed. To compare the different groups at the respective pre-, post-, and retention test, the test results of the specific skills were compared statistically using Kruskal–Wallis tests. The comparison at the time of the pre-test here also represents the basis of the test for homogeneity. Significant results were further statistically compared using pairwise Bonferroni-corrected post hoc Dunn–Bonferroni tests. In addition, the effect size r was calculated for the pairwise post-hoc tests of the Friedman and Kruskal–Wallis tests, respectively. Thereby, 0.1 < r < 0.3 corresponds to a weak effect, 0.3 < r < 0.5 to a medium effect, and r ≥ 0.5 to a strong effect [69]. The *p*-value at which it is considered worthwhile to continue research [70] was set at *p* = 0.05.

## 3. Results

The test results of each skill and the combined values of each test are shown in Figure 2A–D. The results of the statistical analyses are presented in Table 3.

### 3.1. Development within the Groups throughout the Measurement Times

The performance of the DL group improved statistically significantly throughout the course of the measurement from pre- to post- to retention test in the sub-test overhand serve, (*p* = 0.013, r = 0.677), sub-test overhand pass (*p* = 0.001, r = 0.795), and the sub-test underarm pass (*p* = 0.005, r = 0.854) as well as in the combined multiple skills (*p* = 0.000, r = 1.178 ) all with very large effect sizes. The performance level of CI in the overall course developed differently from the order of the DL in all three sub-tests and in the combined multiple skills. The CI group did not reveal a statistically significant difference related to the CO group at post-test in any of the single skills, but showed significant improvement in the overhand serve (*p* = 0.015, r = 0.707) and overhand pass with (*p* = 0.020, r = 0.766) at the retention test with medium to strong effect size but showed no improvement in the underhand pass (*p* = 0.933) as well as in the combined skills (*p* = 0.076).

The performances of the CO group followed the CI group development in the sense that, there was improvement from pre- to post-test but not in the retention test throughout the course of the measurement for the overhand pass sub-test (*p* = 0.004, r = 0.913) and overhand serve (*p* = 0.049, r = 0.677) with medium effect sizes, but not in the underarm pass (*p* = 0.843) as well as in the combined multiple skills (*p* = 0.779).

### 3.2. Development between Groups in the Individual Skills across Measurement Period

An assessment of the groups in the pre-test using Kruskal–Wallis test showed no significant difference in the overhand pass (*p* = 0.867), overhand serve (*p* = 0.926), and underhand pass (*p* = 0.768) as well as in the combined multiple skills (*p* = 0.977).

The Friedman test revealed a statistically significant effect for the groups at the post-test in the overhand pass skill (*p* = 0.013; r = 0.369). The post-hoc pairwise comparison showed the difference to be between CI and DL (*p* = 0.016, r = 0.571) showing a medium to large effect size. There were no statistically significant differences at the retention test in the overhand pass (*p* = 0.164).

In the overhand serve skill, there were statistically significant differences between the groups at post-test (*p* = 0.028; r = 0.296). The post-hoc test showed the difference to be between CI and DL (*p* = 0.023, r = 0.543), representing a medium effect size, and the retention test post-hoc revealed a difference between CO and DL (*p* = 0.023, r = 0.235) accompanied by a weak effect size. The average values show better performances by the DL group than the CI and CO groups in the overhand serve skill.

The underhand pass sub-test revealed a difference between the groups at post-test (*p* = 0.041, r = 0.266) post-hoc test showed the difference between CI and DL (*p* = 0.021, r = 0.470), DL and CO (*p* = 0.041, r = 0.417), all showing a medium effect size.

In the combined multiple skills, a difference in the overhand pass, overhand serve, and underarm pass for only the DL group (*p* = 0.000; r = 1.178) was revealed but no statistically significant difference was found for the CO (*p* = 0.779) and CI (*p* = 0.076) groups, an indication that the DL outperformed CO and CI.

## 4. Discussion

The purpose of this study was to compare the effects of the contextual interference (CI) with the differential learning (DL) applied for stabilizing three volleyball skills (underarm pass, overhand pass, and overhand service) in parallel in advanced beginners in comparison to ball familiarization training (CO).

The nearly constant performances by the control group (CO) in all three skills during the whole study duration from pre- to retention test indicate that improving multiple volleyball skills in parallel seems to need a certain specificity of training even though they were advanced beginners in volleyball. Interestingly, the overhand pass skill showed a statistically significant increase in performance from pre- to post-test. Whether this is caused by a dominance of ball familiarization exercise that are both handed and above head needs to be investigated in detail in future. An aspect that should also be kept in mind is the training effect of the test by itself. The tests contain two above head movements and one underhand movement from below. The greatest intersection of the three skills is the two-handed overhand pass and this would correspond to the logic of DL-theory based on the training of artificial neural nets. This logic would also explain results from previous studies on the CI phenomenon in volleyball. For example, when different serves were practiced exclusively from below [30] or exclusively from above [24] in the acquisition phase, this led to comparable results in each case, which, however, differed from the results in which serves were practiced from below and above [28] in the same phase. From the point of view of ANN research, in the first case [24,30], the neural networks were trained with more “noisy” data for the serve from above only or the serve from below only and were thus more stable against perturbations (from above or from below, respectively) than in the second case [28], in which a network with a larger range of data were trained but much less noisy data were available for the respective areas. To the extent this was supported by both the additional noisy data from the overhand service and the two-handed underhand pass needs to be confirmed by future research. Following the same logic, it is speculated that the combination of the two-handed overhand and underhand pass with the underhand service should lead to larger effects for the underhand pass.

The CI group only reached significant difference to the pre-test as well in the overhand serve. Although the two original CI effects were expressed only in terms of the blocked condition, the data suggest that a random learning sequence of three clearly specified skills generally affects the acute acquisition or stabilization process [2,3]. Whether this is due to the very last training session immediately prior to the post-test or to all random training sessions before requires further investigation. Interestingly again, the two skills that were to play overhand revealed delayed significant improvement at the retention test. Nonetheless, none of the differences related to the CO group reached statistical significance.

A comparison of the results with other studies on CI is problematic because of discrepancies in single or multiple boundary conditions. With an average age of 24 years, the participants in our study were older than all the participants in the other volleyball studies and were the most advanced ones. With 22.5 years [29] and 21.5 years [27] only two studies had participants of comparable age but on a beginners’ level. Both studies designed the investigation with 48 or 72 h for the retention-phase which was shorter than the distance between two subsequent practice sessions. In our case, the retention test followed the post-test after one week which corresponded to the double distance of two subsequent practice sessions. With 1080 learning trials, our study is in the range of [29] with 1215 trials but much larger than the 378 trials in [27]. This is of specific interest as the number of trials was assumed to be influential for the CI effect which in detail should have led to more obvious effects with increasing number of practice trials. All other studies (cf. Table 1) investigated younger participants with much fewer trials for training. Despite comparable ages of the participants in the studies of [27,29], the learning curves differ from ours. Whether this depends on the significantly higher number of trials despite the same number of sessions or whether this depends on the shorter retention phase in the other two studies needs future research. Interestingly, most of the studies on CI either focused only on the combined result of all skills or reported similar time courses for all three skills. Differentiated effects between skills were mainly neglected.

In contrast to the CO and CI groups, the DL group improved in all skills, in post- and retention test, for the average performance. The performance increase from pre- to post-, and to retention test in the DL group coincides with previous studies on DL [61,62,71,72].

From the theory of the CI point of view, it is already challenging to explain the observable phenomena within the CI group because although the subjects are of adequate age and have corresponding playing level, the results differ from previous studies. This becomes even more problematic with respect to explaining the results of the DL group. From CI theory, the enormous increase in variants in DL during the acquisition or stabilization phase combined with the “erroneous” movement executions, which are not allowed in CI theory [7], the results of the DL group should have become worse or at least worse than the performances of the CI group at the post and retention test. This obviously did not happen here and did not happen in other studies on DL, nor in the specific volleyball study of [62].

Accordingly, our findings massively contradict the idea of errorless learning [34,73], for which advocates increased learning rates by avoiding errors. It has not yet been definitively clarified to what extent the associated more positive learning progress is due to the psychological effect of errors or the omission of incorrect movement execution [74,75]. Underlying both cases is the idea of a supposedly correct role model. In the first case, it serves the purpose of comparison and, in case of deviation, leads either to stronger cognitive load due to frustration or through subsequent correction processes. In the second case, it serves the idea of the exclusive inculcation of a correct movement. Both experience a reinterpretation through the DL [5]. Errors are interpreted as fluctuations that are characteristic for living beings and amplified fluctuations are seen as a necessity to make the learning neuronal networks more stable against future disturbances. To what extent replacing judgmental fluctuations (the use of the term error automatically implies knowledge of what is correct, which is rarely available due to the individuality and situatedness of movements) with neutral fluctuations reduces expectation and thus the likelihood of frustration must be shown by future research. Closely related to this is the omission of augmented feedback and corrective instructions from the DL based on the perceived errors. In addition to the detrimental psychological effects, this invariably forces a comparison with either prototypes or memories, which takes up additional cognitive resources and is detrimental to momentary performance. Metaphorically, this process and effect is described by the Chinese proverb “If you want to be unhappy, compare.” Constantly changing movements during DL not only makes it increasingly difficult to remember all the conditions to compare, but also changes the valuation towards deviations.

Furthermore, the CI theory gives neither a statement about the similarity of the exercises that are to be executed in between the repetitions of a to be learned movement nor a statement on the mutual interaction of movements that are acquired in parallel. Although the theory of DL did not explicitly comment on the exact quantitative effects of these topics because of a lack of data, its constant reference to the origin of its name “difference”, the parallel development of diagnostic tools for its quantification [76,77], and the research on the recognition of gross motor patterns of single [78,79] and multiple motor skills [80] as well as on team tactical patterns [81,82] by means of artificial neural nets (ANNs) always implicitly included the importance of similarity for learning. To train an ANN to recognize a single movement technique, it is generally accepted that the recognition rate improves when the ANNs are trained with noisy data [41,83]. When training an ANN simultaneously for multiple movement techniques, the recognition rate depends on the relative similarity of the data for each movement technique. This was recently shown to be exemplarily in a study on pattern recognition of the three throwing disciplines in a decathlon [80]. The greater similarity of discus and shot-put movements for decathletes in the final acceleration phase resulted in higher interdisciplinary recognition rates than those associated with javelin throwing movements. Although these findings are supported by high correlations between shot put and discus throwing performances within a decathlon [84], the extent to which parallel or staggered training promotes or inhibits this process has not yet been clarified. Evidence for similar effects in volleyball is provided by a more detailed examination of the literature data as mentioned earlier.

While the CO and the CI group trained all three skills only with a small amount of noise (CI) or in a nonrelevant area (CO) of the performance relevant neural net, the DL group trained all three skills with increased noise and therefore had a higher chance to force the corresponding neural net to more stability against future disturbances. The “noisiness” of the data in the CI group relative to the CO group is also increased by the random condition of the CI group. By switching between different skills, each skill itself becomes noisier than in a blocked manner, but presumably not noisy enough [85]. Beside the number of exercises and the associated overload of working memory, the idea of overloading the working memory by means of frustration [86] or error processing [33] needs to be kept in mind as well. Interestingly, neither the number of different exercises nor the frustration argument count for the explanation of the DL phenomena. In this context, it is also important to consider that no corrections and no augmented feedback were given in the DL group either, thus not only enabling real self-organization but also reducing the adverse brain activations associated with corrections [87,88]. To fathom the interdependence of increased movement noise and omitting corrections also requires further research.

Overall, the results are more consistent with DL theory than CI theory. DL theory predicts performance gains when training a neural network with noise and assumes the overloading of working memory with multiple parallel gross motor tasks during the exercise phase to produce a qualitative change toward more adequate brain activation to support motor consolidation processes [65,66]. These brain states correspond to those described by the hypo-frontality hypothesis [88], which are often achieved immediately after endurance sports training and lead to effects on executive functions with prolonged use. While endurance exercises normally require more than 15 min of relatively high intensity to achieve these brain states and can rarely be applied multiple times a day, DL training appears to achieve similar brain states after only 3 min [89] and can be applied multiple times a day. In contrast to CI theory, the overloading of the working memory as an explanation for the interference phenomenon after the acquisition phase in CI learning by even more various exercises in DL did not lead to an interfered performance in the post-test but rather led to a qualitative switch in information processing during the processing of multiple gross motor tasks by getting rid of the controlling and limiting activities of the frontal lobe to be able to use additional neuronal resources [63,90]. Limited capacity of working memory also plays a central role in CI theory but is so far undifferentiated related to the involved sensory system [14,36]. Neurophysiological, sensor-specific capacities of working memory are explained in terms of different anatomical structures in the brain for the visual and motor/somatosensory systems [91]. To what extent a shift between sensor-specific working memories occurs during a learning process or to what extent these are individual phenomena still requires extensive research. Tendentially, the literature on the CI-related studies of gross motor movements as a function of age suggests that parallel processing of motor information dominates at the beginning of the learning process and that the focus shifts to the visual aspect (e.g., baseball; volleyball) with increasing age and thereby different capacities of working memory are shaping the learning process.

From a pedagogical cybernetic point of view that is based on Wiener’s [92,93] definition of subjective information and differs from Shannon’s [94] definition of objective information, the constant variation in exercises without repetition in DL learning corresponds on one side to the avoidance of too much redundancy and on the other side to the maintenance of a constant learning rate with constant subjective information [14,95,96]. Thereby subjective information relates to the learning content to the learner’s knowledge and experience, whereas objective information relates to the learning content independent of the learner. Consequently, an identical second repetition of a movement would be completely redundant and would contain no subjective information while the objective information would have the same amount of information all time. Interestingly, the subjective information to be assimilated per unit of time is an individual constant [96]. Because of the evidence of individually constant learning rates from research on cybernetic pedagogy [95,96], the size and structure of noise in learning sequences may be adapted to individuals in the future as well.

## 5. Conclusions

By examining the parallel learning of multiple movements, the present study addressed a previously largely neglected area of motor learning research that is actually ubiquitous in athletic and therapeutic practice. The comparison of the different effects of contextual interference and differential learning chosen for this purpose leads to conclusions for research and practice.

On the research side, the results lead to further specification of existing learning models. Most of the findings in all learning specific groups can be explained by the model of differential learning better than by the contextual interference model. Nonetheless, the detailed discussion raises numerous questions with respect to the similarity of movement, the performance level, the dominant sensory involvement, the duration between the tests relative to the duration between the sessions, and many more. Although the statistics applied do not allow for generalization, the numerous significant results with the corresponding effect sizes may encourage researchers (according to Fisher’s original interpretation of his statistics [70]) to continue the study of differential learning and look for further commonalities and differences to contextual interference learning.

On the practical side, the clear advantages of the DL group confirm once again the predictions originally made [4], especially regarding the improved adaptive capacity due to amplified fluctuations and verify previous study results. The results may support even more coaches and physical education teachers to consider differential learning as originally interpreted [5,97] as an alternative and time-saving approach.

## Figures and Tables

**Figure 1 ijerph-19-10960-f001:**
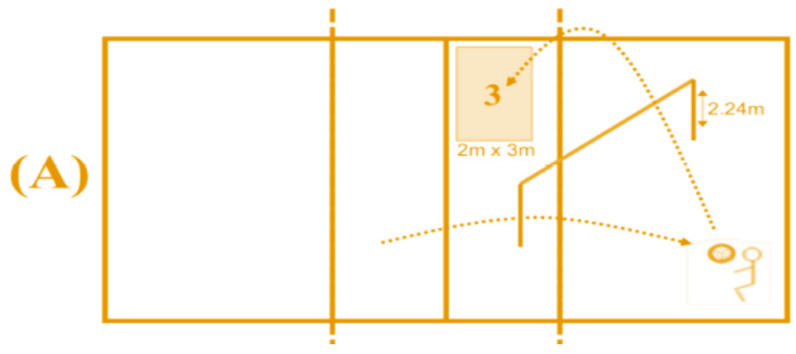
Test designs for the three volleyball skills including scores. Each sub-test corresponds to one skill: (**A**) underhand pass, (**B**) overhand pass, and (**C**) overhand service [48]. The numbers correspond to the scores given when the ball landed in this area.

**Figure 2 ijerph-19-10960-f002:**
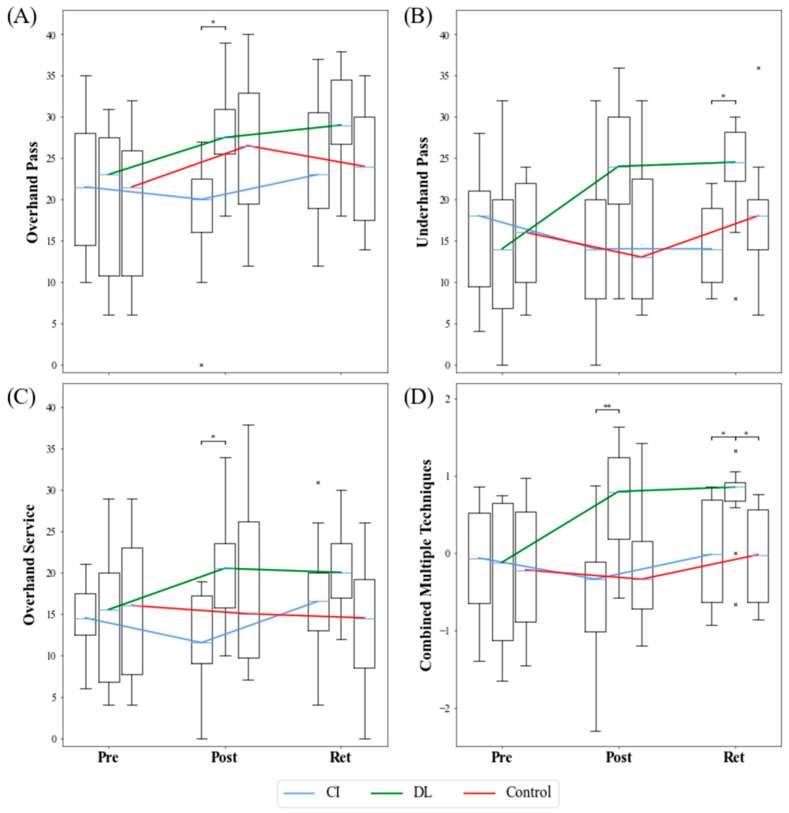
Development of the groups in the test on the respective skills over the duration of the measurement. Values are considered as outliers if they are outside the interval [Q1 − 1.5 × (Q3 − Q1), Q3 + 1.5 × (Q3 − Q1)]. × = outlier (each × stands for one outlier). Brackets show significant differences between the groups only. (* *p* ≤ 0.05; ** *p* ≤ 0.01). Shown are the boxplots of the overhand pass (**A**), underhand pass (**B**), overhand service (**C**), and combined multiple skills (**D**). For the clarity of the development of the groups, the median curves are also shown by line plots. CI = contextual interference group; DL = differential learning group; CO = control group; Pre = pre-test; Post = post-test; Ret = retention test.

**Table 1 ijerph-19-10960-t001:** Studies on stabilizing multiple volleyball skills in parallel within the contextual interference paradigm.

Authors.	Number of Skills to Be Stabilized	Groups	Average Age [Years]	Number of Subjects	Overall Trials	Number of Sessions	Over All Duration [Days]	Post Test [Yes/No]	Retention Test [Hours after Post Test]	Transfer Test [Yes/No]	Post Test: Blocked > Rndm [Yes/No]	Ret./Transf Test: Rndm > Blocked [Yes/No]
Bortolli et al., 1992 [15]	Three/underhand serve/underarm pass/overhand pass	Blocked/random/serial/hi-ser	14.6	53	576	8	60	n	1	y	n	non-systematic
Fialho et al., 2006 [16]	Three serves (tennis, float, Asian)	Blocked/random	16.3	15	184	4	5	y	24	y	n	y
French et al., 1990 [17]	Three/overhead serve/overhand pass/overhand pass	Blocked/random/random-blocked	<14	139	270	9	11	y	48	y	n	n
Jones et al., 2007 [18]	Three/underhand serve/underarm pass/overhand pass	Blocked/random/random-blocked	14	51	270	9	9	y	48	n	n	n
Kalkhoran et al., 2015 [19]	Three/overhead serve/underarm pass/overhand pass	Blocked/random/serial	21.5	60	378	9	9	y	48/72	y	y	y
Meira et al., 2003 [20]	Three serves (underhand, overhead, Asian)	Blocked/random	12.7	36	288	11	11	n	n	y	n	n
Pasand et al., 2016 [21]	Three/underhand serve/underarm pass/overhand pass	Blocked/random/increasing random	22.5	45	1215	9	21	y	48	y	y	y
Travlos 2010 [22]	One serve (underhand–5 directions)	Blocked/random/serial/constant/specific	14.1	72	120	3	21	n	72	y	n	y
Zetou et al., 2007 [23]	Three/?? serve/underarm pass/overhand pass	Blocked/random	12.4	26	270	10	70	y	336	n	n	n

**Table 2 ijerph-19-10960-t002:** Group specific personal data (averages (standard deviations)) of participants.

Group	Age [Years]	Weight [kg]	Height [m]	BMI [kg/m^2^]	Experience [Years]
CI	25.08 (1.62)	68.25 (4.37)	1.68 (0.06)	24.65 (1.77)	7.58 (2.06)
DL	25.83 (2.62)	71.52 (6.78)	1.69 (0.07)	25.15 (2.20)	8.25 (1.76)
CO	23.50 (2.81)	69.08 (3.87)	1.66 (0.07)	25.28 (2.17)	6.83 (2.12)
All	24.81 (2.54)	70.08 (5.18)	1.68 (0.07)	25.03 (2.01)	7.56 (2.02)

**Table 3 ijerph-19-10960-t003:** Statistical comparisons at the three measurement time points within and between groups.

Comparison	Friedman Test and Kruskal–Wallis Test	Post-Hoc Dunn–Bonferroni Tests
OVERHAND PASS		
CO	$\chi^{2}\left(2\right)=11.227,\;$*p* = 0.004 **	Pre vs. Post: *p* = 0.005 **, r = 0.913 +++
Pre-Post-Ret	(Pre: 19.5000, Post: 26.4167, Ret: 23.8333 )	
DL	$\chi^{2}\left(2\right)=13.733,\;$*p* = 0.001 **	Pre vs. Post: *p* = 0.018 *, r = 0.795 +++
Pre-Post-Ret	(Pre: 20.3333, Post: 28.0000, Ret: 29.5833)	Pre vs. Ret: *p* = 0.002 **, r = 0.972 +++
CI	$\chi^{2}\left(2\right)=7.870,\;$*p* = 0.020 *	Post vs. Ret: *p* = 0.024 *, r = 0.766 +++
Pre-Post-Ret	(Pre: 21.3333 Post: 18.0833, Ret: 24.4167)	
Pre	𝜒^2^ (2) = 0.286, *p* = 0.867	
CO-DL-CI	(CO: 17.38 DL: 18.46, CI: 19.67)	
Post	$\chi^{2}\left(2\right)=8.698,\;$*p* = 0.013 *	CI vs. DL: *p* = 0.016 *, r = 0.571 ++
CO-DL-CI	(CO: 20.83, DL: 23.33, CI: 11.33)	
Ret	$\chi^{2}\left(2\right)=3.615,\;$*p* = 0.164	
CO-DL-CI	(CO: 15.63: DL: 23.17, CI: 16.71)	
OVERHAND SERVE		
CO	$\chi^{2}\left(2\right)=6.043,\;$*p* = 0.049 *	Post vs. Ret: *p* = 0.057 *, r = 0.677 ++
Pre-Post-Ret	(Pre:15.9167, Post: 18.2500, Ret: 13.7500)	
DL	$\chi^{2}\left(2\right)=8.619,\;$*p* = 0.013 *	Pre vs. Post: *p* = 0.052 *, r = 0.766 +++
Pre-Post-Ret	(Pre: 15.0833, Post: 20.7500, Ret: 20.3333)	Pre vs. Ret: *p* = 0.008 **, r = 0.559 ++
CI	$\chi^{2}\left(2\right)=8.348,\;$*p* = 0.015 *	Pre vs. Ret: *p* = 0.014 *, r = 0.707 +++
Pre-Post-Ret	(Pre: 14.6667, Post: 11.6667, Ret: 16.9167)	Post vs. Ret: *p* = 0.014 *, r = 0.707 +++
Pre	$\chi^{2}\left(2\right)=0.154,\;$*p* = 0.926	
CO-DL-CI	(CO: 19.46, DL: 18.17, CI: 17.88)	
Post	$\chi^{2}\left(2\right)=7.125,\;$*p* = 0.028 *	CI vs. DL: *p* = 0.023 *, r = 0.543 ++
CO-DL-CI	(CO: 19.00, DL: 23.96, CI: 12.54)	
Ret	$\chi^{2}\left(2\right)=4.349,\;$*p* = 0.114	CO vs. DL: *p* = 0.023 *, r = 0.235 +
CO-DL-CI	(CO: 14.58, DL: 23.38, CI: 17.54)	
UNDERHAND PASS		
CO	$\chi^{2}\left(2\right)=0.341,\;$*p* = 0.843	
Pre-Post-Ret	(Pre: 15.7500 Post: 17.9444, Ret: 18.6667)	
DL	$\chi^{2}\left(2\right)=10.511,\;$*p* = 0.005 **	Pre vs. Post: *p* = 0.003 **, r = 0.854 +++
Pre-Post-Ret	(Pre: 14.7500 Post: 23.6667, Ret: 23.3333)	Pre vs. Ret: *p* = 0.011 *, r = 0.736 +++
CI	$\chi^{2}\left(2\right)=0.140,\;$*p* = 0.933	
Pre-Post-Ret	(Pre: 16.6667, Post: 14.5000, Ret: 14.8333)	
Pre	$\chi^{2}\left(2\right)=0.527,\;$*p* = 0.768	
CO-DL-CI	(CO: 18.75, DL: 16.83, CI: 19.92)	
Post	$\chi^{2}\left(2\right)=6.378,\;$*p* = 0.041 *	CI vs. DL: *p* = 0.021 *, r = 0.470 +
CO-DL-CI	(CO: 15.96, DL: 24.71, CI: 14.83)	DL vs. CO: *p* = 0.041 *, r = 0.417 +
Ret	$\chi^{2}\left(2\right)=9.353,\;$*p* = 0.009 **	CI vs. DL: *p* = 0.003 **, r = 0.601 ++
CO-DL-CI	(CO: 16.46, DL: 25.83, CI: 13.21)	CO vs. DL: *p* = 0.029 *, r = 0.446 +
COMBINED		
CO Pre-post ret	$\chi^{2}\left(2\right)=0.500,\;$*p* = 0.779	
	(PRE: 1.83, POST: 2.08, RET: 2.08)	
DL Pre-post-ret	$\chi^{2}\left(2\right)=18.667,\;$*p* = 0.000 ***	Pre vs. post: *p* = 0.000 ***, r = 1.178 +++
	(PRE: 1.00, POST: 2.67, RET: 2.33)	Pre vs. ret: *p* = 0.001 **, r = 0.942 +++
CI Pre-post-ret	$\chi^{2}\left(2\right)=5.167,\;$*p* = 0.076 * (PRE: 1.92, POST: 1.58 RET: 2.50)	
Pre CO-DL-CI	*χ*^2^ (2) = 0.047, *p* = 0.977 (CO: 18.58, DL: 18.00, CI: 18.92)	
Post CO-DL-CI	*χ*^2^ (2) = 9.664, *p* = 0.008 ** (CO: 16.33; DL: 26.00 CI: 13.17)	CI vs. DL: *p* = 0.009 **, r = 0.609 ++
Ret CO-DL-CI	*χ*^2^ (2) = 10.392, *p* = 0.006 ** (CO:14.25; DL: 26.50, CI: 14.75)	CO vs. DL: *p* = 0.013 *, r = 0.581 ++ CI vs. DL: *p* = 0.019 *, r = 0.557 ++

NB: All *p*-values of the post-hoc tests are Dunn–Bonferroni-corrected. CO = control group; DL = differential learning group; CI = contextual interference group; Pre = pre-test; Post = post-test; Ret = retention test, * *p* ≤ 0.05. ** *p* ≤ 0.01. *** *p* ≤ 0.001. + 0.1 ≤ r < 0.3. ++ 0.3 ≤ r < 0.5. +++ r ≥ 0.5.

## Data Availability

The data that support the findings of this study are available from the corresponding author, WIS, upon reasonable request.

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
