# Peer review of "Learning Multiple Movements in Parallel—Accurately and in Random Order, or Each with Added Noise?"

_ijerph, 2022, doi:10.3390/ijerph191710960_

Round 1
Reviewer 1 Report
The subject matter provided by the authors in comparison with other publications and other published material can be considered original and of interest.
Personally, it is a dense document and not easy to read. I found it tedious. This does not prevent me from congratulating the authors for their work, who state that although the applied statistics do not allow generalization, the numerous significant results with corresponding effect sizes may encourage researchers (according to Fisher's original interpretation of his statistics) to continue the study of differential learning and may support coaches and physical education teachers to consider differential learning in its original interpretation as an alternative and time-efficient approach.
The conclusions are ambiguous, according to the results obtained.
Author Response
Thank you for your encouraging comments. We hope that the tedious character will be reduced at least a little bit by the inclusion of the second reviewers comments which lead to an enormous shortage and removing of repetitions.
Reviewer 2 Report
This is a good study and would be of interest to researchers who do this type of work as well as practitioners who benefit from research in this area. Therefore I recommend the paper as acceptable after minor revisions.
Best regards

Author Response
Many thanks to the reviewer for the valuable comments. The comments made the text more focused and straightforward.
The changes in the text are marked. for the final version the changes just have to be set on "accepted"
- The description of study aim should be redrafted from learning volleyball technique to volleyball skills.
In the whole manuscript the term ‘technique’ has been replaced by ‘skill’.
- That is lack information about results and implications in abstract section (one, two sentences).
The following sentences have been added in the abstract:
“Results showed statistically significant higher rates of improvement in the acquisition and learning phases for the DL group compared to the CI and CO groups. The differences were associated with moderate to high effect sizes in all individual and combined skills. The results show more agreement with DL- than with CI-theory.”
- In my opinion lines 30 to 37 and 161‐162 are not relevant to the manuscript.
Lines 30-37 were deleted. Line 161-162 was rewritten in the following way:
“The differential learning approach (DL), based on system dynamics (SyDy) and artificial neural net (ANN) research [5,44], counterintuitively treated learning as an aspect of living systems that had been largely neglected.”
- In lines 73‐76 the authors indicate that motor skills are separated from technique in the context of manuscript design (pass and serve efficiency). One is directly related to the other in terms of the accuracy of the pass or serve (especially coordination abilities).
The term technique has been replaced by skill in order to avoid the impression of separation. See also comment Nr 1.
- The paragraph starting with line 92 should be moved to the discussion section as support or denial of the results obtained.
In the mentioned paragraph the discussion on similar volleyball skills have been moved to the discussion section. The content on the volleyball studies that was related to the original CI statements were moved on paragraph ahead in order to follow directly to the statement on CI validity.
- The second part of the introduction relating to volleyball from line 211 and below should be redrafted. The introduction in this part of the manuscript contains brief conclusions and constitute the theoretical basis for the presented works for the manuscript. If the authorsʹ aim was to show the methods and results of the presented works in order to refer to their own results, they should be included in the discussion section.
The whole paragraph has been rephrased and shortened.
- In the material and methods section, the table with the data of the respondents should be added (height, weight, BMI and training experience).
A table with height, weight, BMI, and training experience separated by group has been added.
- The second sentence of the intervention section (line 286) is repeated with the design section.
The sentence is deleted
- The description of the intervention in the DL group should be detailed, as it was done in the CI group (Intervention section).
The intervention in the DL group is now described schematically and a reference to the more detailed list in the appendix is provided.
- The same name appears in paragraphs 2.2.1 and 2.2.2, so is it necessary to separate these paragraphs? If so, they should be named differently.
Sorry, this was a formatting error and has been corrected.
- Information on the Multiple Technique design should be added.
The following sentence was added:
“For the multiple skill test the scores of all three single tests were combined by means of a z-transformation.”
- The discussion should be redrafted by removing numerous repetitions from the introduction, results and test design sections.
The discussion was shortened by removing the repetitions from the results and experimental setup sections. However, the repetitions from the introduction have been left there because they have already been removed from the introduction and therefore no longer represent a repetition.
- The application value of the work based on the most important results should be emphasized in the conclusion section.
The conclusions are reformulated.